# TOWARD A MECHANISTIC UNDERSTANDING OF STEPWISE INFERENCE IN TRANSFORMERS: A SYNTHETIC GRAPH NAVIGATION MODEL

## ABSTRACT

Large language models (LLMs) demonstrate impressive capabilities partly due to their use of stepwise inference, such as scratchpad and zero/few-shot chain-of-thought. However, the exact mechanisms and conditions behind these advantages remain unclear. To address this gap, we introduce a synthetic task that is simple, interpretable, and controllable to better understand stepwise inference. Inspired by computational graphs and execution traces, we conceptually analogize the step-wise inference to an autoregressive transformer solving a graph navigation problem on directed acyclic graphs (DAGs). This framework, while simple, allows us to empirically reproduce and quantitatively characterize phenomena observed in LLMs. For example, we demonstrate the superiority of stepwise inference over direct inference and quantitatively characterize a diversity-accuracy tradeoff when the sampling temperature is varied. Having established this foundation, we leverage our synthetic model to reveal new insights into the mechanisms of stepwise inference, such as how a model stitches together sub-paths from the training set to generalize, how a particular graph structure underlying the data generating process affects generalization, and a bias toward shorter paths in inference. Furthermore, in-context chain-of-thought examples can influence the model's navigation, guiding it to follow a given inference path rather than its own potentially biased priors. Overall, this work introduces a grounded synthetic framework for studying step-wise inference and offers mechanistic hypotheses that lay the foundation for a deeper understanding of this phenomenon in LLMs.

## 1 INTRODUCTION

In recent years, Large Language Models (LLMs) (Radford et al., 2018; 2019; Brown et al., 2020) have shown initial "sparks" (Bubeck et al., 2023) of abilities such as reasoning (Huang & Chang, 2022; Webb et al., 2023), mathematical problem-solving (Wei et al., 2022), and planning (Huang et al., 2022), despite being trained solely on the objective of next-token prediction using internet-scale data. The essence of these capabilities is **stepwise inference** (Nye et al., 2021; Wei et al., 2022) and **in-context learning**, first shown by Brown et al. (2020). While numerous compelling observations highlight the benefits of stepwise inference (Nye et al., 2021; Kojima et al., 2022; Wei et al., 2022; Srivastava et al., 2022), the underlying mechanisms of this process remain elusive. *Our goal in this work is to answer the following overarching questions about the mechanisms governing step-wise inference in autoregressive transformer models.*

1. When is stepwise inference more effective than direct inference? How do the properties of the training data matter?

2. How does the model select among multiple possible paths of stepwise solutions? Is there bias?

3. How does sampling temperature affect the accuracy and diversity of model outputs?

4. How can we use prompted context to best control model output?

To address the above questions quantitatively and mechanistically, we formulate a synthetic data generation process that is simple, interpretable, and controllable, while encapsulating key phenomena observed in stepwise inference in LLMs. This will enable development of precise mechanistic hypotheses about how these abilities emerge. Our design philosophy follows the principle of being "as simple as possible, but not simpler." Specifically, our model embodies the following set of properties: (1) the task can be better solved by considering the intermediate steps of computation; (2) there can be several possible paths of computational steps to solve the task; and (3) the context of the task can be controlled by providing exemplars in the prompt.

We argue that graph navigation problems provide a fundamental framework for studying stepwise inference. Graphs give a universal language for modeling and solving complex problems across various domains. Whether it is optimizing network traffic, analyzing social networks, sequencing genetic data, or solving puzzles like the Travelling Salesman Problem, the underlying structure can often be mapped onto a graph (Cormen et al., 2022; Momennejad et al., 2023; Dziri et al., 2023; Saparov & He, 2023). Inspired by algorithmic computational graphs and execution traces, we model stepwise inference as navigating paths in a directed acyclic graph (DAG). Several reasoning problems can be conceptualized in this manner. A chain of logic comprises several elementary logical steps put together in a goal-directed manner and thus involves an element of global planning. In Fig. 1a, the addition of two numbers is decomposed into elementary addition with carry-overs. In Fig. 1b, the Tower of Hanoi game is represented as a graph with nodes corresponding to disc placements on rings, as detailed by (Bubeck et al., 2023). In Fig. 1c, math word problems are represented as chains of logical modules. (See Appendix Fig. 8 for more examples of graph navigation in large-scale LLM studies).

Given a start and goal node, the transformer must autoregressively produce a sequence of nodes that concludes at the goal node. This task requires two levels of computation: locally, each step taken by the model must be valid, and on a global scale, the sequence of steps must be strategically planned in advance to reach the goal node. This setup enables us to control (1) the structure of the underlying graph, (2) the content of the training samples during pre-training, and (3) the information provided to the model in-context before cue the model with the goal and start nodes. Consequently,

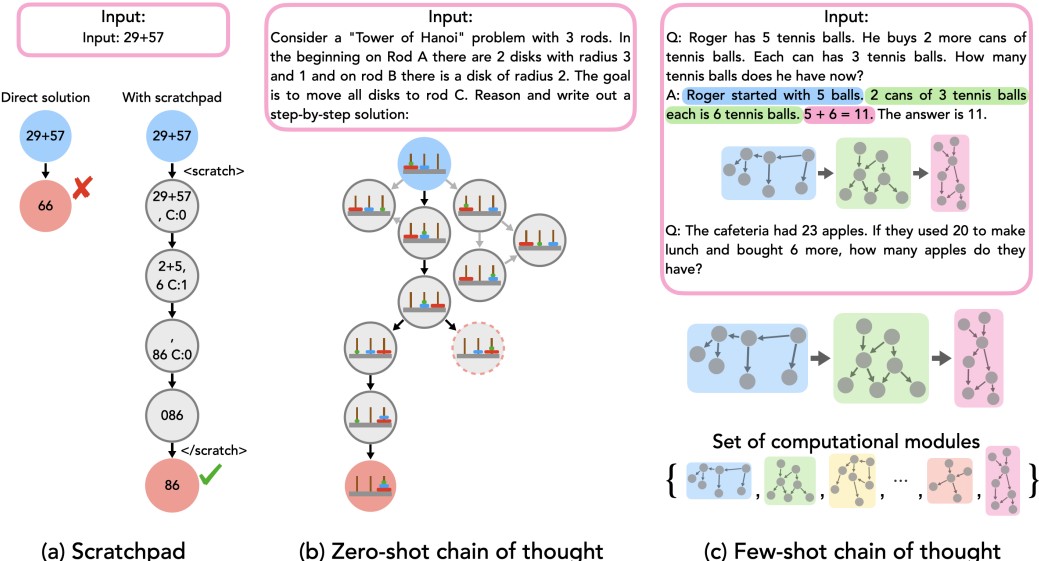

(a) Scratchpad      (b) Zero-shot chain of thought      (c) Few-shot chain of thought

Figure 1: **Graph navigation task as a simple, steerable, and interpretable framework for exploring stepwise inference.** (a) Scratchpad (Nye et al., 2021) improves LLMs' ability to perform complex multi-step computations, such as arithmetic, when they write intermediate computation steps to a buffer called a scratchpad. (b) Zero-shot chain-of-thought prompting (Kojima et al., 2022) improes LLMs' ability to perform multi-step reasoning, such as Tower of Hanoi by prompting them to generate detailed reasoning paths. (c) Few-shot chain-of-thought prompting (Wei et al., 2022) improves LLMs' ability to perform multi-step reasoning, such as solving math word problems (Cobbe et al., 2021), by first presenting an exemplar in-context in the prompt.

we can systematically examine the impact of these properties on the development of reasoning abilities. While (1) and (2) together allows us to explore scratchpad and zero-shot step-by-step reasoning, which is relying on the model's internalized abilities, (3) also delves into few-shot in-context chain of thought prompting (Wei et al., 2022), where predictions are made with guiding examples. Specifically, we examine how in-context exemplars affect the path produced by the model and systematically evaluate the degree of control we have over that path.

In summary, our simplified framework enables experimental study of the phenomenology of stepwise inference and is *simple, steerable, and interpretable*. We harness this framework to make the following contributions:

**Contribution 1. Revealing the Origin of the 'Stepwise Inference Gap'.** In the case of navigating a single fixed underlying DAG in Section 4, we find the existence of a *"stepwise inference gap"* (Prystawski & Goodman, 2023) in the task of determining the path-connectedness of a pair of nodes. When the model is allowed to produce the sequence of nodes step-by-step from start to goal node, it achieves a near-perfect accuracy on determining path-connectedness, as opposed to direct classification without the path. Further, we find that the Step-by-Step Inference gap depends on two key factors, firstly the structure of the underlying DAG: the Step-by-Step Inference gap is larger for graphs that are hierarchical as opposed to random and secondly, the length of the training samples: the Step-by-Step Inference gap is larger if the model has been trained on a set of shorter paths that have to be "stitched" together to build the path during evaluation. We also report other findings: at higher sampling temperatures, the accuracy of the generated path drops exponentially with the length of the path. This effect is a fundamental limitation of autoregressive sampling and has been used to criticize of the use of LLMs for multi-step reasoning. However, with higher temperature, the diversity of paths produced by the model increases a concept we call the "*diversity-accuracy tradeoff*". Further, we find that models have a *shorter path bias* and we study the probability of local and global errors over-training.

**Contribution 2. Exploring the Steerability of Reasoning: *Programming* LLMs with In-Context Exemplars.** *Controllability* is an essential part of reasoning. Approaches to solving the same problem may have differing biases and limitations (Turpin et al., 2023); controllability allows us to choose the most appropriate method for a given context. This flexibility enables adaptation to different scenarios and objectives, ensuring that the solution is both effective and aligned with goals and/or constraints. To examine controllability, in Section 5 we train the model on a task involving a set of DAGs that we refer to as "motifs", inspired by computational primitives that are flexibly recombined and repurposed for different tasks. Here, in-context exemplars chain a subset of these motifs together in a particular order. We find that without any exemplars, the model takes a direct path between a pair of motifs. However, exemplars can be used to steer the path to traverse a set of intermediate motifs. We characterize how the structure and content of these exemplars steer the model's path.

**The motivation, contributions, and limitations of our model-experimental systems approach.** In this work we have adopted the *model-experimental systems approach*, an empirical strategy to precisely characterize and understand smaller, more steerable model systems with the ultimate goal of potentially transferring this understanding to larger-scale complex systems. It is important to clarify the trade-offs and limitations inherent in our approach. Drawing an analogy to the study of biological neural networks, where neural mechanisms identified in small-scale model organisms such as fruit flies or mice may not be directly applicable to medical applications involving the human brain, our observations should not be taken as definitive conclusions directly applicable to large-scale generative models. Instead, our study seeks to establish a minimal synthetic framework, identify data-centric control variables, and formulate mechanistic hypotheses. This lays the groundwork for more in-depth theoretical and empirical investigations of larger models.

## 2 RELATED WORK

There are several puzzling phenomena in the prompts used to elicit chain-of-thought reasoning: chain-of-thought can be improved by sampling methods such as self-consistency (Wang et al., 2022b), prompts might not reflect the true reasoning process used by the language model, as identified by Turpin et al. (2023), the accuracy of the model can be sensitive to the order in which prompts

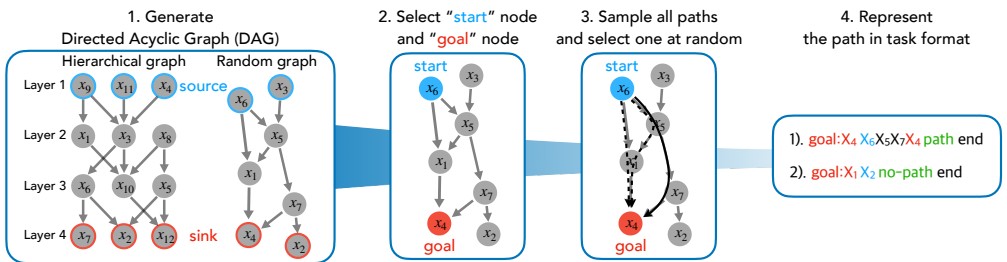

Figure 2: **Data Generating Process for a Single Graph:** This figure illustrates the step-by-step process of generating a training dataset using a single graph. 1) A directed acyclic graph (DAG) is generated, which can be either hierarchically structured or random. 2) A start node and a goal node are selected. 3) All possible paths connecting the start and goal nodes are sampled, and one path is randomly selected. 4) The chosen path is then represented in a task-specific format.

are provided (Lu et al., 2021). Recently, a few works have used theoretical approaches to characterize and explain chain-of-thought. Li et al. (2023) studies in-context learning of random MLPs and finds that a transformer that outputs the values of intermediate hidden layers achieves better generalization, Feng et al. (2023) shows that with stepwise reasoning, transformers can solve dynamic programming problems, and Prystawski & Goodman (2023) studies reasoning traces in transformers trained to learn the conditionals of a Bayes network. At large scale, there have been several efforts to explore the phenomenology of chain of thought. Saparov & He (2023) introduce a synthetic dataset called PrOntoQA to systematically study the failure modes of chain of thought in the GPT3 family fine-tuned on the dataset and find that misleading steps of reasoning are the most common cause of failure in the best-performing models. Chen et al. (2023) find that chain-of-thought fails at compositional generalization and counterfactual reasoning. Dziri et al. (2023) formally study how LLMs solve multi-step reasoning tasks and find that these models fail at true compositional reasoning and reduce most multi-step reasoning tasks to linearized sub-graph matching, essentially learning 'short-cut solutions' (Liu et al., 2022) also called 'rules of thumb' (Madirolas et al., 2023). Momennejad et al. (2023) study in-context graph navigation in LLMs, finding that they fail to do precise planning. Wang et al. (2022a); Schaeffer et al. (2023) find that the content of the exemplars is less relevant to accuracy than their syntactic structure. Razeghi et al. (2022) find that the accuracy of reasoning is correlated with the frequencies of occurrence in the pretraining dataset. For an extended discussion of related work, see Appendix A.1.

## 3 DEFINING A SYNTHETIC GRAPH NAVIGATION TASK

We use **directed acyclic graphs (DAGs)** to study steps of inference. DAGs are a natural mathematical abstraction to study formal reasoning chains: as described in Dziri et al. (2023), the output of any deterministic algorithm can be represented as a DAG. For example, in programming languages, a computational primitive is the smallest 'unit of processing' that a program may use. Every program in this language can be written as a sequence of primitives chained together in series. To be concise, we repeat the construction of Dziri et al. (2023): let $A$ be any deterministic algorithm and $\mathcal{F}_A$ be the set of computational primitives used by $A$. Given a set of inputs to the algorithm $\mathbf{x}$, $G_{A(\mathbf{x})}$ is its *computational graph*, defined as follows. Let $V$ be the set of variables involved in the algorithm $A$, each taking values in set $s$. Let $E$ be the set of edges, i.e., function arguments in an intermediate computation. Node $v$'s parents are $\mathrm{pa}(v)$. Therefore, for some primitive $f \in \mathcal{F}_A$, $f(v) = f(\mathrm{pa}(v))$. Thus, $G_{A(\mathbf{x})} = (V, E, s, \mathcal{F}_A)$ and the *source nodes* of $G_A$ represent the inputs $\mathbf{x}$ to the algorithm and the *sink nodes* represent the output of the algorithm $A(\mathbf{x})$ [1].

Yet another motivation comes from linguistics and natural language syntax (Chomsky, 2002). Every sentence in a language can broken down into its syntactic or parse tree, which is a special case of a directed acyclic graph. For example, the sentence 'I drive a car to my college' can be parsed as the following graph: ('I': Noun phrase, 'drive a car to my college': Verb Phrase) → ('drive': Verb, 'a

---

[1]Sink nodes are all nodes $X$ s.t. $\mathrm{children}(X) = \emptyset$ and source nodes are nodes $X$ s.t. $\mathrm{parents}(X) = \emptyset$

car': Noun Phrase, 'to my college': Prepositional Phrase) → ('a': Determiner, 'car': Noun), ('to': Preposition, 'my college': Noun Phrase) → ('my': Determiner, 'college': Noun).

## 3.1 PRELIMINARIES

A DAG **G = (N,E)** is made up of set of nodes $N = \{X_i\}_{i=1}^{|N|}$ and set of directed edges across the nodes $E = \{(X_i, X_j)\}_{X_i, X_j \in N}$. The edges of a DAG are captured by its **adjacency matrix** $A$ where $A_{ij} = 1$ if $(X_i, X_j) \in E$.

A **directed simple path** is a sequence of distinct nodes of **G** which are joined by a sequence of distinct edges. The first node of a path is referred to as the **start node** and the last node is the **goal node** (Fig. 2).

**Structure of the DAG** To create a feedforward hierarchical DAG we construct a set of $L$ layers with $N$ nodes each. For every node $n_l$ in layer $l$ and $n_{l+1}$ in layer $l + 1$, we draw a directed edge $(n_l, n_{l+1})$ with probability $p$, which we refer to as **edge density**. Thus on average, between any two layers there are $pN^2$ edges and each node in an intermediate layer has an out-degree and in-degree of $pN$. The number of paths from a particular node in layer $l$ to layer $l' > l$ is exponential and given by $(pN)^{l'-l}$ - this is quantified in the path length distribution shown in Appendix Fig. 9. Lastly, the generated graphs contain no disconnected components. The nodes from layer 1 are the **source nodes:** nodes $\{X_i\}$ of DAG **G** with parents$(X_i) = \emptyset$ and the nodes from layer L are **sink nodes:** nodes $\{X_i\}$ of DAG **G** with children$(X_i) = \emptyset$.

To create[2] a random DAG of $N$ nodes, we first create a random upper triangular adjacency matrix $A_{N \times N}$ with bernoulli entries with edge density $p$, such that $p(A_{ij} = 1) = p$. We also ensure that the graph is connected. This results in a bell-shaped path length distribution (Appendix Fig. 9). For a more detailed discussion of construction and sampling from the DAGs, refer Appendix A.3.

When the DAG is hierarchical, between a start and goal node, nodes in the intermediate layers must be visited (Appendix Fig 9) whereas when the DAG is random, there is a uniform probability that 2 nodes are connected and there is no explicit notion of hierarchy.

In both these scenarios, we can define the notion of **path diversity:** between any 2 path-connected nodes, there can be several possible paths. We quantify the path diversity in random and hierarchical graphs in Fig. SI Fig 9.

## 3.2 DATA GENERATING PROCESS FOR SINGLE GRAPH SCENARIOS

We focus on two setups in this work, where *one allows for context and one does not*. This is intentional so that we can explicitly analyze benefits of stepwise inference in the presence of extraneous context, which may influence a model's internalized knowledge, and hence its execution.

**Single graph scenario.** Zero-shot chain-of-thought (Kojima et al., 2022), planning (Huang et al., 2022) or scratchpad prompting (Nye et al., 2021) are paradigms that describe how simple prompts such as "let's think step-by-step" allow an LLM to produce intermediate steps and show improved accuracy in reasoning tasks. We model these paradigms in our task using a single underlying DAG and include a special token at the start of the sequence which signifies that the model has to produce all intermediate steps.

**Prompt structure and training data generation** In the single graph setting with underlying DAG $G$, each prompt is made from a single simple path. Given a start node $X_{\text{start}}$ and goal node $X_{\text{goal}}$, the model has to classify whether there exists a path from $X_{\text{start}}$ to $X_{\text{goal}}$. We create a pair of tokens `path` and `no-path`. We constructed two datasets: one that contains stepwise inference and another that does not. Examples of prompts are provided below. For stepwise inference, the path between the start node $X_4$ to the goal node $X_6$ is represented as `goal` : `X_4` `X_6` $X_5$ $X_7$ `X_6` `path` `end`. Without stepwise inference, the example path is represented as `goal` : `X_4` `X_6` `path` `end`.

## 4 RESULTS ON SINGLE-GRAPH SCENARIOS

Our first result is that a transformer trained on directed edges and a small fraction of node pairs from a fixed underlying DAG can generalize to *all* node pairs, including those held out during training, producing valid simple paths from start to goal nodes (see appendix for details). Thus the model can

---

[2]See Appendix A.3 for algorithms for generating graphs as well as their path statistics

Figure 3: **Advantage of Stepwise Inference in Graph Navigation Tasks:** (a) In random graphs, stepwise inference shows an advantage over direct inference in connectivity prediction tasks. (b) This advantage is further pronounced in hierarchical graphs, where the distances between nodes can be significantly larger. (c) We show that the stepwise inference gap arises when the training set contains paths that are shorter than the paths required to connect nodes in the evaluation set. (d) This indicates that stepwise inference is beneficial when a model must trim and connect paths it has learned during training to generalize effectively: The red, green, and blue paths are subsets of paths seen during pretraining while the combined path is one produced by the model during evaluation.

'stitch' or mix-and-match (sub)paths it has observed during training to produce a valid path across a pair of held-out connected nodes.

**A single underlying graph: The stepwise inference gap** Findings from large-scale experiments have indicated that the inclusion of intermediate reasoning steps results in increased accuracy when solving a stepwise inference task (Kojima et al., 2022). Owing to stepwise nature of our task, we hypothesize that similar phenomena will occur: to classify a given start and goal node pair as path or no-path, the model has two modes of operation: either produce the whole path from the start to the goal or directly classify. We hypothesize that a model trained in the former manner will have higher classification accuracy.

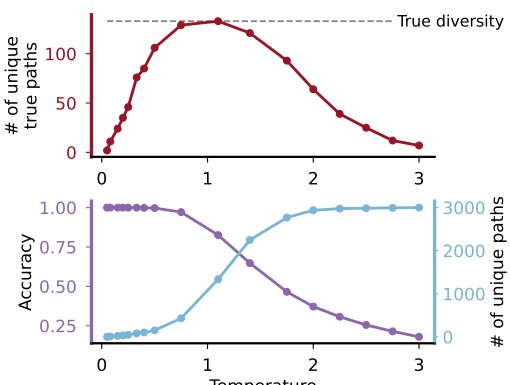

Figure 4: **A diversity vs. accuracy trade-off in finite temperature stepwise inference for transformers:** As sampling temperature is increased, the diversity of paths generated by the model from a single $(n_{start}, n_{goal})$ pair increases, while the accuracy of the path decreases. This tradeoff is captured by measuring the number of unique *true* paths which is non-monotonic (top), showing the existence of an optimal temperature for sampling. The dashed line denotes the ground truth path diversity of $(n_{start}, n_{goal})$.

Fig. 3 shows the accuracy of path/no-path classification for (a) a random DAG and (b) a hierarchical DAG. We trained two distinct models using two types of datasets: one with stepwise inference paths and one without. We find that the model trained on the dataset with stepwise inference (represented by the blue line) achieves higher classification accuracy than the model without stepwise inference (the pink line) in both cases. We refer to the difference in classification performance with and without stepwise inference as the 'stepwise inference gap'. We also observe that the stepwise inference gap is larger for hierarchical graph than for random graph. Appendix Fig. 11 shows that our results hold even when tokens are randomly corrupted to mimic noisy real world data.

**Stitching of paths**

Further, we hypothesize that stepwise inference is useful when the training data has the following structure: (1) the underlying DAG is hierarchical, which means that there is an explicit feed-forward ordering of nodes and to go from nodes in one layer to next one must pass through all intermediate layers and (2) the model must 'stitch' together subsets of paths seen over pretraining in flexible ways to generalize. To test this, we trained the model using paths from hierarchical DAGs while varying the lengths of paths in the training data. Specifically, we created training data that contains start nodes from layer $l$ and goal nodes from layer $l'$ and restricted $l' - l < \Delta$, where $\Delta$ denotes the length of the path. During evaluation, we choose node pairs such that $l' - l \geq \Delta$. Since the

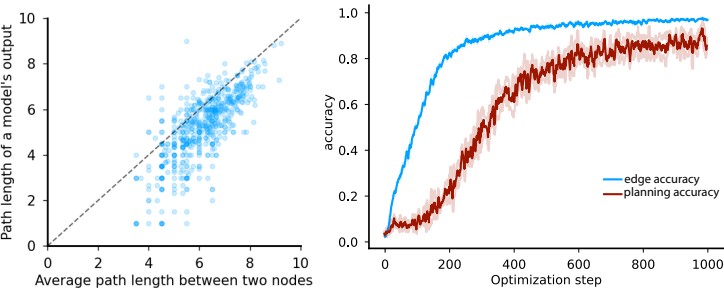

Figure 5: (a) **Model outputs are biased toward shorter paths.** We compare the average lengths of actual and model-generated paths in a random graph, revealing the model's bias toward generating shorter paths. (b) **The learning dynamics of failure mode probabilities over training:** It can be seen that the model first learns to produce correct edges (effectively bigram statistics) and then learns the global objective of producing a path that ends at the cued goal node. Accuracy curves are averaged over 3 trained models with different random seed.

number of simple paths for a hierarchical graph increase exponentially with path length (number of paths $\approx (pN)^{l'-l}$, SI Fig.9), for smaller values of $\Delta$, the model has observed a much smaller number of paths. Thus, the model must combine and piece together several different paths seen over pretraining to effectively solve the task. In Fig. 3(c), we observe that the smaller the value of $\Delta$ used during pretraining, the greater the stepwise inference gap becomes. This is because the shorter the paths seen during training, the more recombination the model has to do – we hypothesize that this is where intermediate steps and scratchpads will most improve accuracy

**The diversity-accuracy tradeoff with higher sampling temperatures**

LLMs rely on sampling for next-token generation. At low temperatures, this process is deterministic but to get a variety of responses, higher temperatures are necessary. However, at higher temperatures the model is more likely to make mistakes or 'hallucinate'. Systematically understanding and calibrating this diversity-accuracy tradeoff (Zhang et al., 2020) is crucial for tailoring the behavior of generative models to specific tasks and desired outcomes. Fig. 4 illustrates the effect of sampling temperature on the accuracy and diversity of the generated paths. LLM inference at 0 sampling temperature is equivalent to taking the most likely token at each time step (the maximum likelihood estimate). In this setting, the model deterministically generates the same path for any given provided pair of start and goal nodes: $n_{\text{start}}$ and $n_{\text{goal}}$. However, in the underlying graph, there are typically numerous paths from each $n_{\text{start}}$ to $n_{\text{goal}}$. To capture this diversity, we fixed the start node $n_{\text{start}}$ and the goal node $n_{\text{goal}}$, and prompted the model 3,000 times, sweeping through different sampling temperatures in Fig. 4. Here, we observe a trade-off, which we term the ***diversity-accuracy tradeoff***: At lower sampling temperatures, the model produces fewer paths, all of which are accurate and true (as shown by the purple line in the bottom panel). Conversely, as the sampling temperature rises, the paths become more diverse (the blue line) but less accurate (the purple line). Accuracy is defined as the probability that the path both has true edges (i.e., no missteps) and ends at the provided $n_{\text{goal}}$ while diversity is the number of unique paths generated (the blue line in the bottom panel). To the best of our knowledge, this phenomena has not been quantitatively studied before.

**A bias towards shorter paths:**
Fig. 5a examines the average path lengths in a random graph, comparing true paths to those generated by our trained model. Notably, the model consistently produces paths that, on average, are shorter than the actual paths in the random graph. This observation suggests that the model has a *bias towards efficiency*, which can lead to oversimplification of complex stepwise inference or omission of important intermediate steps, similar to 'shortcut solutions' (Liu et al., 2022).

**Failure modes of step-by-step inference:**
Given underlying DAG $G$, during step-by-step inference, the model produces a sequence of nodes from the start node $n_{\text{start}}$ which must terminate at the goal node $n_{\text{goal}}$, given by the sequence $n_0 = n_{\text{start}} \to n_1 \to n_2 \to ... \to n_k \to ... \to n_T$. Here in our setup, there are two broad categories of failures possible (Saparov & He, 2023):
**Misstep:** $(n_k, n_{k+1}) \notin G$. An edge produced by the model does not exist in the DAG.

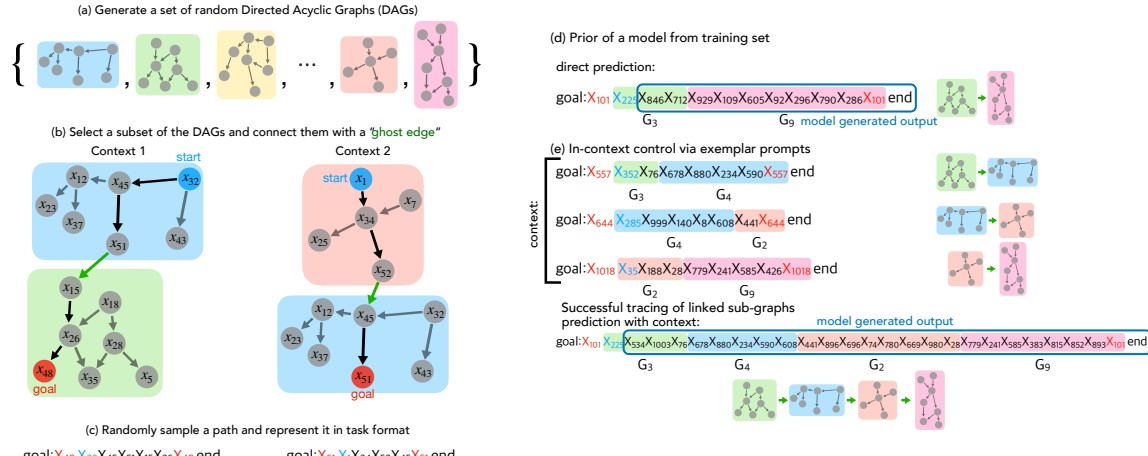

Figure 6: **Data Generating Process for Connected Sub-Graphs (Motifs):** This figure illustrates the step-by-step process of generating a training dataset by combining multiple subgraphs (motifs). (a) We start by making a set of random directed acyclic graphs (DAGs). (b) Next, we pick a subset of these DAGs and connect them together using "ghost edges" to create a bigger graph. (c) From this bigger graph, we randomly sample paths and turn them into a task format. **Example output sequences from the model highlighting the steerability of stepwise inference.** (d) Direction prediction: Given $n_{\text{start}} \in G_3$ and $n_{\text{goal}} \in G_9$, the model produces a path from $G_3 \to G_9$, placing a single ghost edge $(X_{712}, X_{929})$. (e) With in-context exemplars: primitive sequences from $G_3 \to G_4$, $G_4 \to G_2$ and $G_2 \to G_9$ in-context make the model steer its navigation through the path stringing together these motifs in order: $G_3 \to G_4 \to G_2 \to G_9$, placing a ghost edge between every consecutive motif, for a total 3 ghost edges.

**Planning failure:** $n_T \neq n_{\text{goal}}$. The model produces a path that does not terminate at the goal. Learning dynamics of these failures modes are presented in Fig. 5b and suggest that the capability of global planning emerges after the model has learned to take correct steps.

## 5 RESULTS ON MULTI-GRAPH SCENARIOS

The single graph setting let us explore *zero-shot* planning and stepwise reasoning, where the model relied purely on knowledge internalized over pretraining for stepwise planning. To study how context can influence model's path, we introduce the concepts of motifs and in-context exemplar paths.

### 5.1 DATA GENERATING PROCESS FOR THE MULTI-GRAPH SCENARIO

To model few-exemplar based chain-of-thought prompting, we modify our single graph setup to include a set of subgraphs that we refer to as **motifs**, denoted by $\{G_i\}_{i=1}^n$. A motif is a DAG that is fixed across pretraining and inference. Before we describe the construction of the in-context examples, we will define a few terms: (i) **Ghost edge:** For a pair of connected motifs $G_i \mapsto G_j$, an edge between a sink node of $G_i$ and a source node of $G_j$, and (ii) A **primitive sequence** (Fig. 6) is sequence of nodes across 2 motifs $G_i$ and $G_j$ with a start node in $G_i$ and goal node in $G_j$ and this sequence contains exactly 1 ghost edge. Fig. 6. In chain-of-thought prompting (Wei et al., 2022), one or more examples of reasoning are provided before asking the next question, as illustrated in Figure 1(c). The LLM then generate a chain-of-thought which matches that of the exemplar. To model this, we chain a subset of $k$ motifs $G_{c_1} \to G_{c_2} \to ... \to G_{c_k}$ together and provide exemplars. Each exemplar $e$ is a primitive sequence across each pair of consecutive motifs: $e \in (G_{c_i} \to G_{c_{i+1}})$ which contains exactly 1 ghost edge. The construction of a primitive sequence is described in Fig. 6 and examples are shown in Fig. 6(c). Given a start node $n_{\text{start}} \in G_{c_1}$ and a goal node $n_{\text{goal}} \in G_{c_K}$, the model can be prompted either directly (Fig. 6(d)) or provided with exemplars and then queried for a path from $n_{\text{start}}$ to $n_{\text{goal}}$ (Fig. 6(e)).

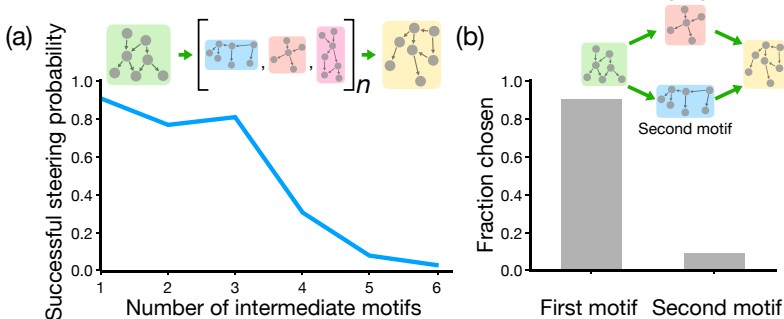

Figure 7: **How do the number of examplers affect the controllability of motifs?** (a) As we vary the number of intermediate motifs in a chain, the path generated by the model follows the path described by the chain until $n = 4$, which is the extent of the training data. (b) In the case of 2 conflicting chains in-context, the model has a bias to pick the first chain.

## 5.2 THE STEERING OF NAVIGATION BY EXEMPLARS

We find that the model can successfully follow the chain defined by the in-context exemplars. An example output produced by the model is in Fig.6 (right), highlighting the path the model takes through the chain of motifs $G_3 \rightarrow G_4 \rightarrow G_2 \rightarrow G_9$. We also find that the model generalizes to arbitrary orders of motifs strung out, including those that did not occur consecutively in the training data – in other words, in-context control is capable of *compositional generalization* (Li et al., 2023).

## 5.3 HOW DO THE EXEMPLARS AFFECT CONTROLLABILITY OF GRAPH NAVIGATION?

Next, we study how the structural content of the exemplars affects the navigation path chosen by the model. We hope to shed some light on and create hypotheses for the vast and varied findings about stepwise reasoning in LLMs at scale.

**Number of intermediate motifs:** In Fig.7(a), we varied the number of exemplars provided to the model. This is equivalent to stringing together a longer chain of motifs to navigate over. We find that the model can generalize well to unseen orders of motif up to the maximum number chained together in the training data. We hypothesize that for chain-of-thought and related methods at scale: *the model will fail to generalize to reasoning chains longer than those present in its training data.*

**Bias towards the first exemplar in the case of conflict:** Multiple examples of context provided in the prompt can increase the precision of our control over the model, but it can also lead to confusion. Here, we systematically and quantitatively study the behavior of the model when two contexts are provided but are in conflict. In Fig. 7(b) To model scenarios with conflicting exemplars, we study a case where two chains of motifs are provided, starting from the same set of primary motifs and ending at the terminal motif. We find that the model has a strong bias toward choosing the first chain over the second. This result is qualitatively similar to what happens at scale with large context windows (Liu et al. (2023)).

## 6 CONCLUSION

Our grounded synthetic task gives researchers control of variables that are generally not practical to control in the real-world. These "knobs" encompass a variety of parameters, such as the structure of the underlying graph (be it random or hierarchical), the extent of training data paths, and inference temperature. Furthermore, in the in-context setting, we can manipulate the content, sequence, and volume of exemplars, among other factors. This framework stands as a unique playground or laboratory, though with its limitations, presenting insights into what stepwise inference is capable of and where its limitations lie.

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

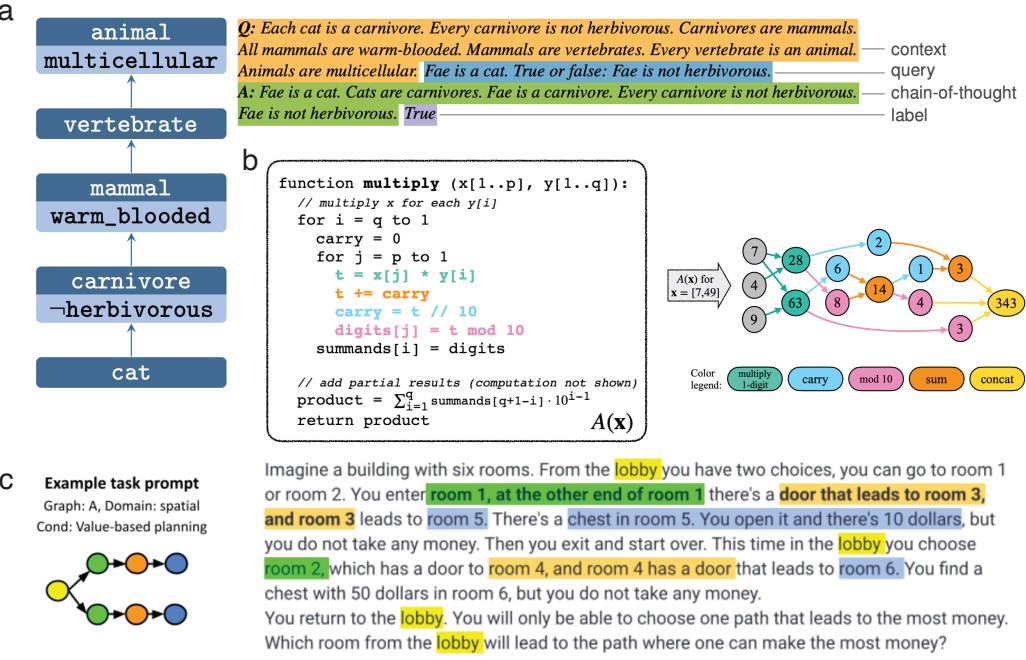

Figure 8: **Examples of stepwise inference as graph navigation in LLM evaluations:** [Figures taken from respective papers] (a) An example graph created for a prompt (left) from the ProntoQ&A dataset (Saparov & He, 2023) (b) (Dziri et al., 2023) studies how simple algorithms such as multiplication of digits can be represented as a graph (c) CogEval (Momennejad et al., 2023) studies many large scale LLMs such as ChatGPT-4 and Claude2 on planning and navigation tasks.

## A  APPENDIX

### A.1  WHY GRAPH NAVIGATION?

In this section we will elaborate on our paradigm of graph navigation to study stepwise.

- Saparov & He (2023) Study simple linear DAGs as models of first order logical reasoning. They construct *ontologies* Fig. 8a and prompt LLMs to do analogical reasoning.

- Dziri et al. (2023) study mathematical expression evaluation in large scale LLMs as DAG navigation Fig. 8b. Any mathematical expression can be decomposed into elementary computations which are chained together.

- Momennejad et al. (2023) evaluates many large scale LLMs such as ChatGPT-4 and Claude2 on synthetically designed planning and navigation tasks Fig. 8c.

- Allen-Zhu & Li (2023) studies transformers trained on context-free grammars (CFGs) which are DAGs.

Large language models (LLMs) have been shown to possess sophisticated and human-like reasoning and problem-solving abilities (Srivastava et al., 2022). Chain-of-thought or scratchpad reasoning refers to many similar and related phenomena involving multiple intermediate steps of reasoning *generated internally and autoregressively* by the language model. First described by Nye et al. (2021); Kojima et al. (2022), adding prompts such as 'think step by step' allows the LLM to autonomously generate intermediate steps of reasoning and computation, improving accuracy and quality of its responses. This is referred to as zero-shot chain-of-thought. A related set of phenomena, few-shot chain-of-thought prompting (Wei et al., 2022) occurs when the language model is shown exemplars of reasoning before being prompted with a reasoning task. The model follows the structure of logic in these exemplars, solving the task with higher accuracy.

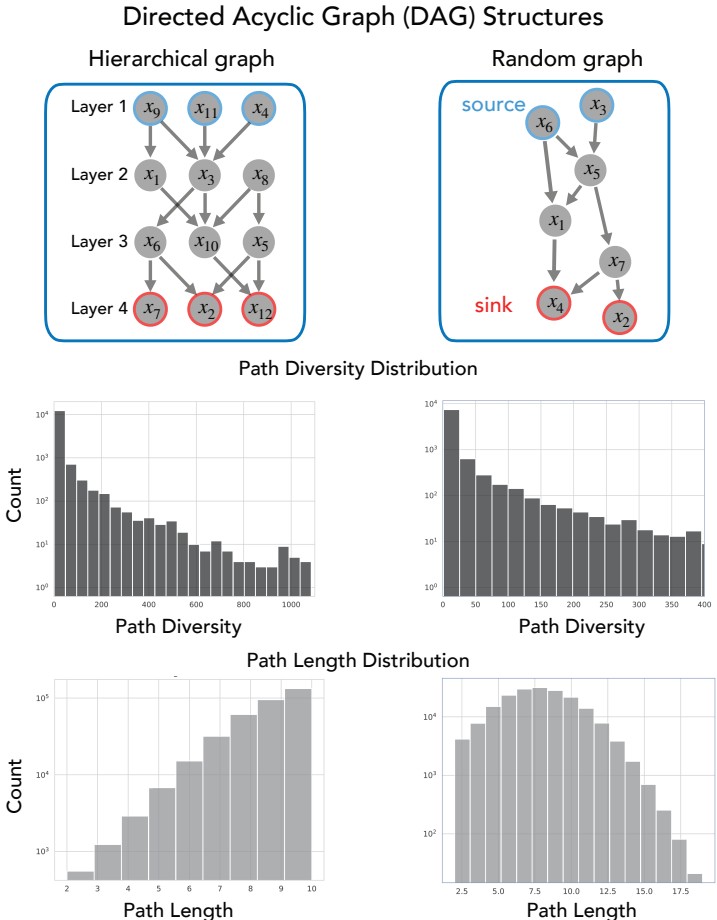

Figure 9: **Construction and properties of Hierarchical and Random DAGs:**(top) Schematic of hierarchical and random graphs. Hierarchical graphs are organized into layers with connections only between nodes of successive layers but random graphs have no such structure. (middle) Path diversity is defined as the number of paths between any 2 path connected nodes. (bottom) Path length distributions: Owing to the hierarchical nature, the path length distribution is exponential in hierarchical graphs where it is more Gaussian-like for randomm graphs.

There have been several prompting strategies developed, all of which rely on sampling intermediate steps: tree-of-thoughts (Yao et al., 2023), graph-of-thoughts (Besta et al., 2023), program-of-thoughts (Chen et al., 2022) and methods which use more than 1 LLM: such as STaR (Zelikman et al., 2022), RAP (Hao et al., 2023), Selection-Inference (SI) (Creswell et al., 2022; Creswell & Shanahan, 2022).

### A.2 SETUP AND CONSTRUCTION OF GRAPH AND MODEL

Here we describe the properties of the DAGs we use, the training setup, model architecture and hyperparameters.

We use 2 DAG structures, hierarchical and random (Fig. 9). Random DAGs are constructed by randomly generating an upper-triangular matrix where each entry has probability $p$ of existing. Hierarchical DAGs are generated by predefining L sets of nodes and drawing an edge between a node $n_l$ in layer $l$ and $n_{l+1}$ in layer $l + 1$ with probability $p$. Lastly, we ensure that the graph is connected. These lead to different path diversity and path length distributions, which affect the efficacy of stepwise inference, as shown in our results.

For training, we tokenize every node and we use next-token prediction with a cross entropy loss:

$$\mathcal{L}(\mathbf{x}_n, \text{target } n) = -\log\left(\frac{\exp(\beta x_{n, \text{ target } n})}{\sum_{t=0}^{\#\text{tokens}} \exp(\beta x_{n,t})}\right) = -\log\left(\underbrace{\text{softmax}(\beta \mathbf{x}_n)_{\text{target } n}}_{\textbf{prob}(\text{target } n)}\right) \quad (1)$$

| Hyperparameter | Value |
|---|---|
| learning rate | $10^{-4}$ |
| Batch size | 64 |
| Context length | 32 |
| Optimizer | Adam |
| Momentum | 0.9, 0.95 |
| Activation function | GeLU |
| Number of blocks | 2 |
| Embedding dimension | 64 |

Table 1: Hyperparameters of the transformer

For model architecture, we use a GPT based decode-only transformer with a causal self-attention mask. Our implementation is based on the popular nanoGPT repository[3].

Each transformer block contains a causal attention layer, layer-norms, residual connections and an MLP (see Fig. 10). The MLP contains two fully-connected layers sandwiched by a GELU layer (Hendrycks & Gimpel, 2016).

The input tokens are converted to one-hot vectors before being passed through to the Transformer. The model makes use of no dropout and no biases in the Layer norm layers. We use weight-

---

[3]available at https://github.com/karpathy/nanoGPT

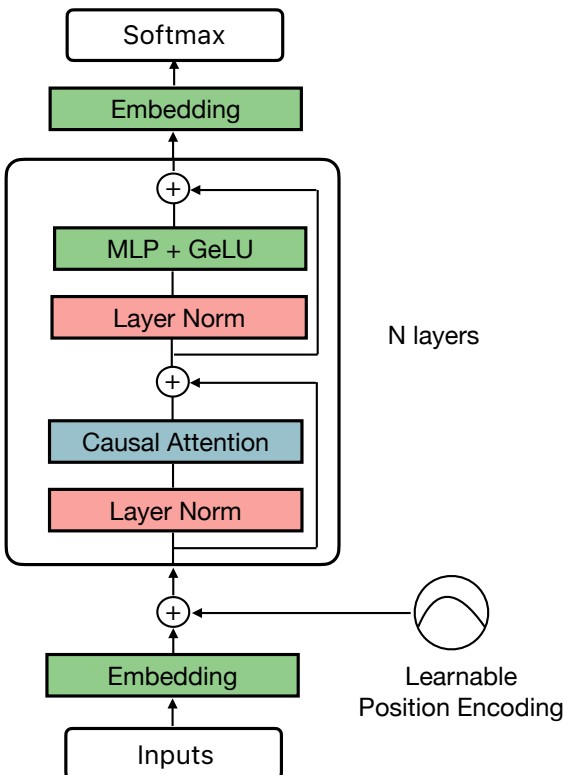

Figure 10: **The architecture of GPT (Radford et al., 2019) style decode-only transformers**

tying (Press & Wolf, 2016) in the Transformer which uses shared weights for the input and the output embedding layers. Finally, we make use of mixed-precision (bf16 in torch) to speedup training.

### A.3 TRAINING PROTOCOL FOR EXPERIMENTS

---

**Algorithm 1** Generate Random connected DAG

---

nodeNames ← ['X' + str(i) for i in range(numNodes)]
**function** CREATEUPPERTRIANGULARMASK(n, p)
    matrix ← random binary matrix with size $n \times n$ and probability $p$ for 1s
    upperTriangular ← extract upper triangular part of matrix
    **return** upperTriangular
**end function**
**repeat**
    adjMatrix ← CREATEUPPERTRIANGULARMASK(numNodes, p)
    dag ← create directed graph in NetworkX from adjMatrix and nodeNames
**until** dag is connected

---

**Algorithm 2** Generate Hierarchical Connected Random DAG

---

p ← [probability of connection between layers]
nodesPerLayer ← [number of nodes in each layer]
numLayers ← [total number of layers]
numNodes ← nodesPerLayer × numLayers
**function** CREATELAYEREDDAG(nodesPerLayer, numLayers, p)
    Initialize an empty directed graph G in NetworkX
    **for** currentLayer ← 1 **to** numLayers − 1 **do**
        **for** each node $j$ in currentLayer **do**
            **for** each node $k$ in currentLayer + 1 **do**
                **if** random number ≤ p **then**
                    Add edge from node $X_j$ to node $X_k$ in G
                **end if**
            **end for**
        **end for**
    **end for**
    **return** G
**end function**
**repeat**
    dag ← CREATELAYEREDDAG(nodesPerLayer, numLayers, p)
**until** dag is connected

---

For single graph experiments, we randomly generate either a hierarchical graph or a random graph G with $N = 200$ nodes. In the random graph setting the probability of an edge $p = 0.05$ while in the hierarchical graph, the probability of an edge between a node in layer $l$ and layer $l + 1$ is $p = 0.05$, and we choose 10 layer with 20 nodes each.

**Test-train split:** To generate training data correspond to path connected node pairs, we first put all edges (which are paths of length 1) into the training data and further, we generate all simple paths between every pair of nodes in G and put all paths corresponding to 20% of nodes into the training data, while the remainder are held out evaluations.

For the non-path connected pairs, we simply take all node pairs apart from the ones which have simple paths between them and add 5000 of these node pairs into the training data, chosen to roughly balance the classes.

For each node pair, we use the prompt format described in the main text:

For stepwise inference goal : $X_4$ $X_6$ $X_5$ $X_7$ $X_6$ path end . For direct prediction:
goal : $X_4$ $X_6$ path end .

## A.4 ADDITIONAL EXPERIMENTAL RESULTS

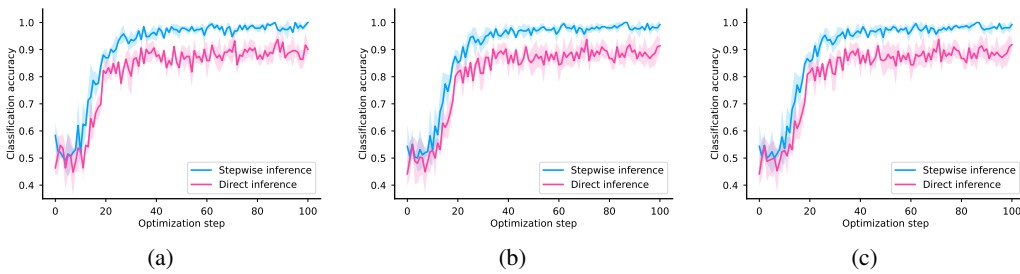

(a)          (b)          (c)

Figure 11: **Stepwise inference gap with corrupted tokens:** In this experiment, (a) 5%, (b) 10% and (c) 20% of tokens were randomly corrupted to mimic real world language data. The stepwise-inference gap persists.

In Fig. 11, we mimic real-world language data, abundant in ambiguity and polysemy, by corrupting (a) 5%, (b) 10% and (c) 20% of tokens in a single graph scenario. To achieve this, we replaced a randomly chosen 5% and 10% of the tokens in the training data with random tokens. We observe that the gap between stepwise inference and direct inference persists in both scenarios. This finding indicates that stepwise inference remains effective in more realistic settings with noise.

In Fig. 12, we swept the density of the graph from 0.08 to 0.12 on a hierarchical graph. We observe a stepwise inference gap in all cases. The stepwise inference gap becomes smaller for larger densities.

Fig. 13 presents a density plot comparing the average lengths of actual paths with those generated by the model in a random graph. This observation verifies the model tends to produce shorter paths between a given pair of start and goal nodes.

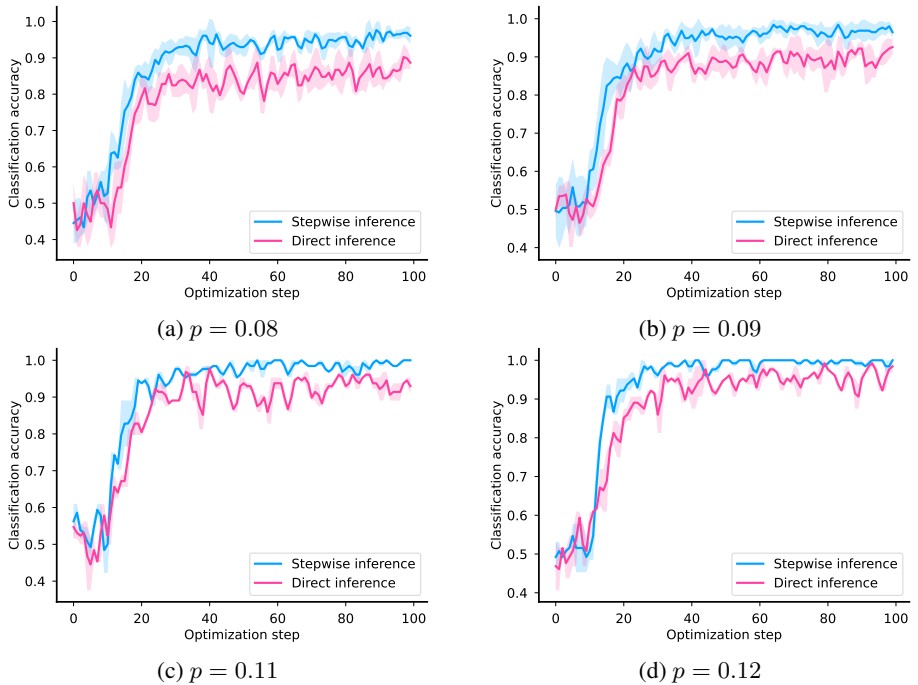

(a) $p = 0.08$

(b) $p = 0.09$

(c) $p = 0.11$

(d) $p = 0.12$

Figure 12: **Advantage of Stepwise Inference in Graph Navigation Task.** Here we vary p, the edge density of connectivity in the graph.

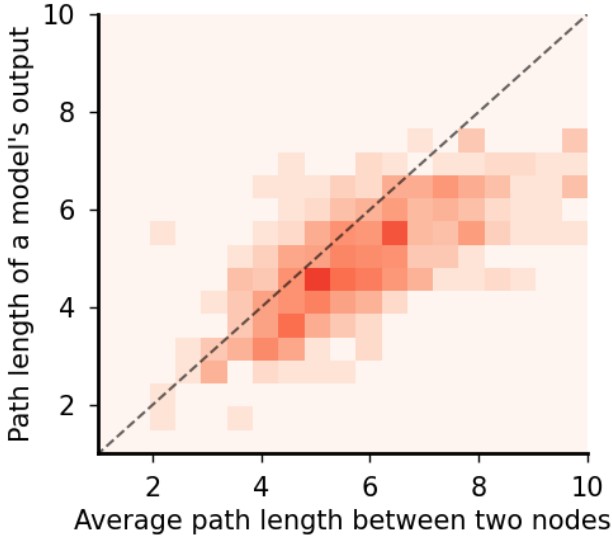

Figure 13: **Model outputs are biased toward shorter paths.**

