# OpenReview forum: "Toward a Mechanistic Understanding of Stepwise Inference in Transformers: A Synthetic Graph Navigation Model"
_ICLR.cc/2024/Conference — Submitted to ICLR 2024_

### Official Review · Reviewer_V1TB · 2023-10-31

**Soundness:** 3 good
**Presentation:** 3 good
**Contribution:** 3 good
**Rating:** 6
**Confidence:** 2

**Summary:**

This paper sheds light on step-wise inference in LLMs by casting the problem as a graph navigation problem. A dataset of random DAGs is generated with start and goal nodes and an autoregressive transformer is trained to navigate the graph from start to goal by next step prediction. The problem can be mapped to step-wise inference in LLMs and the paper investigates several settings in which this is manifested.

**Strengths:**

I have found the problem setting of the paper interesting - these are important questions and some of the more challenging aspects of LLMs are investigated here.

The casting to DAG navigation is a good choice - it covers a lot of potentially related inference problems with a simple model which is easy to generate data for and train.

**Weaknesses:**

I think my main issue with the paper is that LLMs are trained on very very different data (though in a roughly similar setup).
I am not sure the mapping between DAG navigation and what LLMs actually learn is as simple as claimed in the paper as LLMs need to do many other things when trained.

**Questions:**

Would the authors shed more light on the limitations and dissimilarities between the proposed model and actual LLM training?

---

> ### Author Response · Authors · 2023-11-20
> **Author Response part 1**
>
> We thank you for your positive feedback and we are happy that you think casting stepwise inference as graph navigation is a good choice and that our setting was interesting.
> We also realized that we might not have motivated the connection between graph navigation and stepwise inference. We have now made it more explicit and also added examples from LLMs (Appendix Figure 8)
>
> >**Comment:** I think my main issue with the paper is that LLMs are trained on very very different data (though in a roughly similar setup). I am not sure the mapping between DAG navigation and what LLMs actually learn is as simple as claimed in the paper as LLMs need to do many other things when trained.
>
> **Response:** This is a valid concern and we have significantly improved the writeup to address it further. To emphasize, we note that our use of graph navigation was meant as an abstraction of reasoning tasks where stepwise inference protocols have been shown to substantially improve performance. While we did not intend to suggest that graph navigation is a universal model for all such tasks, we do believe that for several tasks currently studied in literature [1,2,3,4], it serves as a faithful abstraction. For example, a LLM is often used to solve problems like the following (taken from [1]): “Imagine a building with 6 rooms. From the lobby you have two choices, you can go to room 1 and room 2. You enter room 1 and at the end of room 1 there is a door that leads to room 3, and room 3 leads to room 5. There’s a chest in room 5. You open it and there is 10 dollars but you do not take the money. Then you exit and start over. This time in the lobby you choose room 2, which has a door to room 4, and room 4 has a door that leads to room 6. You find a chest with 60 dollars in room 6 but you do not take the money. You return to the lobby. You will be able to choose 1 path that leads to the most money. Which room from the lobby will lead to the path where one can make the most money.” *Implicitly*, the computation required to solve this task involves graph navigation, though the graph is latent – the model has to map natural language onto a graph. This is similar to our setup, where a path sampled from a graph serves the same purpose: it represents a reasoning task that involves achieving a goal (navigation to a node), but the underlying graph is *implicit* and never fully observed. One can similarly consider the example of math word problems (from GSM8K [5]): “A carnival snack booth made 50 dollars selling popcorn each day. It made three times as much selling cotton candy. For a 5-day activity, the booth has to pay 30 dollars rent and 75 dollars for the cost of the ingredients. How much did the booth earn for 5 days after paying the rent and the cost of ingredients?” Implicitly, the model has to parse these natural language sentences into a computational graph and navigate on it.
>
> Given the similarities highlighted above between our setup and practical tasks, we do believe the claims discussed in our work will transfer to real world settings. However, to perform validation of our generated hypotheses, such as the diversity-accuracy tradeoff and short path bias in stepwise inference, we believe a dedicated study is needed and hence leave it for future work.

---

> > ### Author Response · Authors · 2023-11-20
> > **Author Response part 2**
> >
> > We also note that to ensure the point above is addressed in the paper, we have substantially revised our manuscript to better motivate the explicit connection between graph navigation and stepwise inference in more realistic settings. Specifically, we have rewritten parts of the introduction (we have elaborated our exposition on the relation between stepwise inference and graph navigation), rewritten our abstract, contextualized our results more thoroughly with existing literature , and also edited the formatting and arrangement of figures and results in paper (including promoting the training dynamics figure to the main text since it shows that global planning capability arises \text{after} models learn to take correct steps). As an example, our edits now highlight several recent studies that either use or model stepwise inference with LLMs to study tasks that are specific instantiations of our proposed graph navigation problem (e.g., see [1, 2, 3, 4]; we have also added Appendix figure 8 which shows examples of graph navigation from these recent LLM evaluation studies). Given that these studies exist in the (very recent) literature, we argue the connection between stepwise inference, planning, and graph navigation is explicit and our updated draft ensures this connection is emphasized upon.
> >
> >  We have also added a limitations section (at the end of our introduction) to acknowledge the fact that our proposed task is an abstraction, which the reviewer rightfully pointed out.
> >
> > In summary, we hope that we have addressed the reviewer's concerns and that the reviewer can adjust their score to reflect this fact and support our paper for accceptance.
> >
> > [1] Mommenejad et al. Evaluating Cognitive Maps in Large Language Models with CogEval: No Emergent Planning (NeurIPS 2023)
> >
> > [2] Saparov et al. Language Models Are Greedy Reasoners: A Systematic Formal Analysis of Chain-of-Thought (ICLR 2023)
> >
> > [3] Dziri et al. Faith and Fate: Limits of Transformers on Compositionality (NeurIPS 2023)
> >
> > [4] Hou et al. Towards a Mechanistic Interpretation of Multi-Step Reasoning Capabilities of Language Models. (arXiv, 2023)
> >
> > [5] Cobbe et al. Training Verifiers to Solve Math Word Problems (2021) https://arxiv.org/pdf/2110.14168v2.pdf

---

### Official Review · Reviewer_JQZB · 2023-11-01

**Soundness:** 3 good
**Presentation:** 3 good
**Contribution:** 4 excellent
**Rating:** 6
**Confidence:** 3

**Summary:**

To analyze the stepwise inference in large language models, the authors examine a synthetic graph navigation problem and replicate several phenomena originally observed in LLMs. This work provides a simplified platform for researchers to study properties of LLM.

**Strengths:**

1. A comprehensive framework of DAG is provided as substitute for stepwise inference in LLM
2. Reasoning gap, the diversity-accuracy tradeoff, in-context control are replicated in well-designed experiments, showing the effectiveness of the proposed enviroment.
3. Well-written, easy to follow.

**Weaknesses:**

1.While several phenomena originally observed in LLMs were replicated in this DAG couterpart. It will be more convincing if some extraoplated property in DAG system can be found in LLMs which was not discovered before.

2. Emergent abilities of large language lodels were thought to be a consequence of unpredictable scaling, therefore, I doubt if a simpilified DAG substitute can replicate this behaviour.

**Questions:**

I will increase my score if my concern is well addressed.

---

> ### Author Response · Authors · 2023-11-20
> **Author Response part 1**
>
> We thank you for your review and enthusiasm about our work! We are happy to hear that you think our framework is comprehensive, our experiments were well-designed and that our paper was well-written. We respond to specific comments below.
>
> >**Comment:** While several phenomena originally observed in LLMs were replicated in this DAG couterpart. It will be more convincing if some extrapplated property in DAG system can be found in LLMs which was not discovered before.
>
> **Response:** We appreciate the reviewer’s concern that we might have only replicated phenomena at scale. We would like to take the opportunity to clarify that our study does more than mere replication of phenomena already observed in LLMs.
>
> Firstly, we ground and validate our synthetic setup by replicating phenomena observed at large scale, namely the step-wise inference gap.
>
> Next, we go beyond by:
> (i) identifying new phenomena leveraging the fact that we have an interpretable ground truth in the data-generating process. Specifically, we have found and characterized the following phenomena:
>
> - The dependence of the stepwise inference gap on the structure of the underlying graph
>
> - A short-path bias when there exist several paths between a pair of nodes
>
> - The diversity-accuracy tradeoff
>
> - The emergence of global planning capability *after* learning to take correct steps over training
>
> - The dependence on the length of the context during training to generalization during evaluation
>
> - The model’s bias towards specific examples when there exist conflicting examples in context
>
>  (ii) We have been quantitative and precise while characterizing these phenomena. Leveraging the interpretability, steerability that our controlled synthetic setup provides, we have access to ground truth and thus we can put exact numbers on the model’s behavior. Evaluation of LLMs is a looming open problem [1], with no existing solution due to the scale, complexity, and lack of accessibility of their huge training corpus. Data leakage is a confounding issue which makes evaluation inherently challenging due to the ambiguity of defining test-train split and similar ambiguity for defining appropriate metrics [2]. Our synthetic setup allows us to bypass both these issues.
>
> In principle, a further study could be performed by training models at scale and confirming at scale, which is what the reviewer is suggesting. We have acknowledged this limitation of our current work in a new section at the end of our introduction. This is a fundamental trade-off when one decides to use synthetic data: we gain interpretability and steerability by having a synthetic DGP where everything is controlled and this allows us to make precise quantitative statements and formulate hypotheses for the origin of phenomena.
>
> Furthermore, we’d like to point the reviewer to a few studies where graph navigation was tested in-context in language models at large-scale:
>
> Mommenejad et al. Evaluating Cognitive Maps in Large Language Models with CogEval: No Emergent Planning (NeurIPS 2023)
>
> Saparov et al. Language Models Are Greedy Reasoners: A Systematic Formal Analysis of Chain-of-Thought (ICLR 2023)
>
> Dziri et al. Faith and Fate: Limits of Transformers on Compositionality (NeurIPS 2023)
>
> We have included a figure in our appendix containing example prompts and model outputs from each of these papers.
>
> Moreover, a recent mechanistic study even designs probes to extract the underlying logical graph during stepwise inference in LLMs!:
>
> Towards a Mechanistic Interpretation of Multi-Step Reasoning Capabilities of Language Models Hou et al (2023)  https://arxiv.org/pdf/2310.14491.pdf

---

> > ### Author Response · Authors · 2023-11-20
> > **Author Response part 2**
> >
> > We have also communicated the goals of our work and our reasons for taking a synthetic approach in our global response.
> >
> > As another example of the benefits of studying a synthetic setup, consider the phenomenon of grokking [3] was first discovered in a toy modular addition task. This led to a spur of research on mechanistic interpretability and feature learning [4,5,6,7,8,9,10,11,12] which revealed several novel deep learning phenomena. None of the mentioned works performed experiments in large-scale models, but were nonetheless extremely valuable. We view our work as falling into this category.
> > Additionally, performing experiments with real LLMs will require thorough considerations, experiment design that is difficult, if not impossible, to do carefully in the time given for the rebuttal period and resources – we thus leave this for future work.
> >
> > [1] Chang et al. A Survey on Evaluation of Large Language Models https://arxiv.org/abs/2307.03109
> >
> > [2] Schaeffer et al. Are Emergent Abilities of Large Language Models a Mirage? (NeurIPS 2023)
> >
> > [3] Power et al. Grokking: Generalization Beyond Overfitting on Small Algorithmic Datasets (2022) https://arxiv.org/abs/2201.02177
> >
> > [4] Liu et al Towards Understanding Grokking: An Effective Theory of Representation Learning (NeurIPS 2022)
> >
> > [5] Liu et al Omnigrok: Grokking Beyond Algorithmic Data (2022) https://arxiv.org/abs/2210.01117
> >
> > [6] Kumar et al Grokking as the Transition from Lazy to Rich Training Dynamics (2023) https://arxiv.org/abs/2310.06110
> >
> > [7]  Thilak et al (2022) The Slingshot Mechanism: An Empirical Study of Adaptive Optimizers and the Grokking Phenomenon
> >
> > [8] Zhong et al The Clock and the Pizza: Two Stories in Mechanistic Explanation of Neural Networks https://arxiv.org/abs/2306.17844
> >
> > [9] Liu et al Grokking as Compression: A Nonlinear Complexity Perspective https://arxiv.org/abs/2310.05918
> >
> > [10] Gromov.  Grokking modular arithmetic (2023) https://arxiv.org/abs/2301.02679
> >
> > [11] Nanda et al Progress measures for grokking via mechanistic interpretability https://arxiv.org/abs/2301.05217 (2023)
> >
> > [12] Nanda, Neel. The Case for Analyzing Toy Language Models. LessWrong Forum, 28th December 2022
> > https://www.lesswrong.com/posts/GWCgZrzWCZCuzGktv/200-cop-in-mi-the-case-for-analysing-toy-language-models

---

> ### Author Response · Authors · 2023-11-20
> **Author Response part 3**
>
> >**Comment:** Emergent abilities of large language models were thought to be a consequence of unpredictable scaling, therefore, I doubt if a simpilified DAG substitute can replicate this behaviour.
>
>
> **Response:** We *respectfully* disagree with the claim that emergent abilities only occur in large scale models. Abilities like in-context learning (ICL) and stepwise inference protocols like chain-of-thought (CoT) have been shown and extensively studied in small toy models by several recent works by designing well defined synthetic tasks [1,2,3,4,5,6,7,8,9,10]. The goal of such studies (and ours well) is to study interesting phenomenology of LLMs and design abstract setups that enable development of precise mechanistic hypotheses for how the phenomenon works. For example [4,5] use a synthetically designed setup to develop hypotheses for how ``knowledge” about an entity is stored in a pretrained model, showing that such knowledge can be manipulated via relatively simple linear transformations. This development of mechanistic hypotheses is precisely what we have done with our experiments.
>
> We also note that to ensure the point about relation to LLMs is addressed in the paper, we have substantially revised our manuscript to further motivate the explicit connection between graph navigation and stepwise inference in more realistic settings. Specifically, we have rewritten parts of the introduction (we have elaborated our exposition on the relation between stepwise inference and graph navigation, added a discussion of the limitations of a synthetic data approach), rewritten our abstract, contextualized our results more thoroughly with existing literature (added Appendix figure 8 which shows examples of graph navigation in a LLM), and also edited the formatting and arrangement of figures and results in paper (including promoting the training dynamics figure to the main text since it shows that global planning capability arises *after* models learn to take correct steps). As an example, our edits now highlight several recent studies that either use or model stepwise inference with LLMs to study tasks that are specific instantiations of our proposed graph navigation problem (e.g., see [11,12, 13, 14]). We have included a figure in Appendix (Fig. 8) that describes example prompts and model outputs from each of these papers to exemplify the relation of our proposed synthetic benchmark to realistic setups and tasks considered by prior work in showing the value of stepwise inference. Given that these studies exist in the (very recent) literature, we argue the connection between stepwise inference, planning, and graph navigation is explicit and our updated draft ensures this connection is emphasized upon.
>
> With our detailed responses and changes to the paper, we hope that we have addressed the reviewer's concerns and the reviewer increases their score.
>
> [1] Bai et al. Transformers as Statisticians: Provable In-Context Learning with In-Context Algorithm Selection (NeurIPS 2023)
>
> [2] Akyurek et al. What learning algorithm is in-context learning? Investigations with linear models (2022) https://arxiv.org/abs/2211.15661
>
> [3] Ahn et al. Transformers learn to implement preconditioned gradient descent for in-context learning (NeurIPS 2023)
>
> [4] Zeyuan Allen-Zhu and Yuanzhi Li, (2023) Physics of Language Models: Part 3.1, Knowledge Storage (2023)
>
> [5] Zeyuan Allen-Zhu and Yuanzhi Li, Physics of Language Models: Part 3.2, Knowledge Manipulation (2023)
>
> [6] Maya Okawa et al. Compositional Abilities Emerge Multiplicatively: Exploring Diffusion Models on a Synthetic Task (NeurIPS 2023)
>
> [7] Lindner et al (2023) Tracr: Compiled Transformers as a Laboratory for Interpretability https://arxiv.org/abs/2301.05062
>
> [8] Feng et al, Towards demystifying the mystery behind chain of thought: A theoretical perspective (2023)
>
> [9] Li et al. Dissecting Chain-of-Thought: Compositionality through In-Context Filtering and Learning (NeurIPS 2023)
>
> [10] Prystawski et al. Why think step by step? Reasoning emerges from the locality of experience (NeurIPS 2023)
>
> [11] Mommenejad et al. Evaluating Cognitive Maps in Large Language Models with CogEval: No Emergent Planning (NeurIPS 2023)
>
> [12] Saparov et al. Language Models Are Greedy Reasoners: A Systematic Formal Analysis of Chain-of-Thought (ICLR 2023)
>
> [13] Dziri et al. Faith and Fate: Limits of Transformers on Compositionality (NeurIPS 2023)
>
> [14]  Hou et al (2023) Towards a Mechanistic Interpretation of Multi-Step Reasoning Capabilities of Language Models  https://arxiv.org/pdf/2310.14491.pdf

---

> > ### Comment · Reviewer_JQZB · 2023-11-22
> > **Reply to rebuttal**
> >
> > Thank you for the detailed rebuttal. Most of my concerns are well addressed. i recommend this paper for publication.

---

> > > ### Author Response · Authors · 2023-11-22
> > > **Adjusting score to reflect your new opinion about our paper**
> > >
> > > Dear Reviewer,
> > > Thank you for acknowledging that
> > >
> > > > Most of my concerns are well addressed
> > >
> > > in our recent revision. We are pleased to have met your expectations in addressing the key issues you raised.
> > >
> > > In light of your earlier statement:
> > >
> > > > I will increase my score if my concern is well addressed
> > >
> > > we kindly request you to revisit your score of our paper. An updated score reflecting your current opinion would greatly assist in the fair and accurate assessment of our work.
> > > Thank you for your consideration.

---

### Official Review · Reviewer_3PuG · 2023-11-01

**Soundness:** 2 fair
**Presentation:** 2 fair
**Contribution:** 2 fair
**Rating:** 3
**Confidence:** 4

**Summary:**

The paper proposes to study the step-wise inference mechanism by exploring the graph navigation problem.

**Strengths:**

+ Originality:

    The idea of modeling step-wise inference as graph navigation is interesting.

**Weaknesses:**

- Quality:

    i) Although the authors claim to reveal the connection between graph navigation and the step-wise inference mechanism in transformer, the experiments, to my understanding, are solely about training transformers on the synthesized graph dataset. It remains unclear to me how this set-up can be translated into the study of step-wise inference mechanism, despite the conceptual similarity between DAG and the step-wise inference mechanism. It is unclear whether this finding on the synthesized graph dataset can be safely transferred to real-world large-scale dataset.

    ii) The definition of hierarchical graph and random graph is confusing to me. It seems possible to convert the random graph in fig. 10 into a hierarchical graph by simply re-grouping the nodes.

- Significance:

    The highlighted finding, i.e., the step-wise inference gap is influenced by (i) the underlying DAG structure, and (ii) the length of the training samples is only studied in a very shallow level. The DAG structure in the context seems to be not well-defined and well-categorized. The discussion about the length of the training samples mainly focuses on how it affects the length of the model output.

**Questions:**

Please see the weaknesses section.

---

> ### Author Response · Authors · 2023-11-20
> **Author response Part 1**
>
> We thank you for your review and glad to hear that you find our idea of modeling step-wise inference as graph navigation interesting. We address your specific concerns below.
>
> >**Comment:** Although the authors claim to reveal the connection between graph navigation and the step-wise inference mechanism in transformer, the experiments, to my understanding, are solely about training transformers on the synthesized graph dataset. It remains unclear to me how this set-up can be translated into the study of step-wise inference mechanism, despite the conceptual similarity between DAG and the step-wise inference mechanism. It is unclear whether this finding on the synthesized graph dataset can be safely transferred to real-world large-scale dataset.
>
> **Response:** Thank you for this question, which touches on very important aspects of the motivation, objectives, and limitations of our work. Your question has prompted us to add the new paragraph at the end of the Introduction on "The motivation, contributions, and limitations of our model-experimental systems approach." (Please also see our global response for related discussion as well.) To provide further evidence for the correctness of the analogy between our synthetic graph navigation task and stepwise inference tasks that real LLMs are evaluated on, we have also created a new Appendix Figure 8, where we showcase several recent works [1,2,3,4] that have studied the impact of stepwise inference on tasks that can easily be modeled in our graph navigation framework. We elaborate key arguments below.
>
> First, our contributions can be divided into three main parts of (i) formulation of synthetic graph navigation model motivated by the analogy between DAG and the step-wise inference; (ii) empirical validation of the model by precise reproduction of phenomena observed in LLMs; and (iii) discovery of new mechanistic hypotheses such as the impact of the underlying graph structure on stepwise inference, path stitching mechanisms, diversity-accuracy tradeoff, shorter-path bias in inference, and more.
>
> In response to your specific concerns, beyond the "conceptual similarity between DAG and the stepwise inference mechanism", we have experimentally validated our model by reproducing an array of phenomena observed in LLMs. Furthermore, while all experimental results are obtained by "training transformers on the synthesized graph dataset", as you correctly point out, we believe that our novel mechanistic hypotheses are our important contribution. Finally, we agree with the necessary trade-offs and limitations of our study that our proposed hypotheses are not conclusive about the mechanisms underlying LLMs. We have clarified this in our new paragraph as, "To draw an analogy to the study of biological neural networks, where neural mechanisms identified in small-scale model organisms such as fruit flies or mice may not be directly applicable to medical applications involving the human brain, our observations should not be taken as definitive conclusions that are readily applicable to modern large-scale generative models.” Nevertheless, we are hopeful that our novel hypotheses can translate to larger-scale settings, based on the fact that our model system, despite its simplicity, successfully reproduced an array of known phenomena in LLMs. Testing them would require training and evaluating models on larger-scale setup, but would be an exciting future direction.

---

> ### Author Response · Authors · 2023-11-20
> **Author response part 2**
>
> To summarize, the goal of our paper was to use graph navigation as an abstraction of tasks where stepwise inference is known to play a critical role and correspondingly develop and validate precise hypotheses that can help understand stepwise inference. This is similar to several recent works studying synthetic problems for developing a better understanding of LLMs [4,5, 6, 7, 8,9,10]. As we note above, we believe our abstraction is faithful and hence we expect our results will transfer to realistic scenarios. Such an investigation is best left to future work, however. We also emphasize that to further clarify the comments above, we have rewritten parts of the introduction where we have elaborated our exposition on the relation between stepwise inference and graph navigation, rewritten our abstract and also edited the formatting and arrangement of figures and results in paper (including promoting the training dynamics figure the main text since it shows that global planning capability arises *after* models learn to take correct steps).
> We hope this has clarified the contributions and limitations of our work, but we are happy to discuss further if you have any remaining concerns.
>
> [1] Mommenejad et al. Evaluating Cognitive Maps in Large Language Models with CogEval: No Emergent Planning (NeurIPS 2023)
>
> [2] Saparov et al. Language Models Are Greedy Reasoners: A Systematic Formal Analysis of Chain-of-Thought (ICLR 2023)
>
> [3] Dziri et al. Faith and Fate: Limits of Transformers on Compositionality (NeurIPS 2023)
>
> [4] Hou et al. Towards a Mechanistic Interpretation of Multi-Step Reasoning Capabilities of Language Models. (arXiv, 2023)
>
> [5] Zeyuan Allen-Zhu and Yuanzhi Li, (2023) Physics of Language Models: Part 3.1, Knowledge Storage (2023)
>
> [6] Zeyuan Allen-Zhu and Yuanzhi Li, Physics of Language Models: Part 3.2, Knowledge Manipulation (2023)
>
> [7] Maya Okawa et al. Compositional Abilities Emerge Multiplicatively: Exploring Diffusion Models on a Synthetic Task (NeurIPS 2023)
>
> [8] Zhou et al. What Algorithms can Transformers Learn? A Study in Length Generalization. (arXiv, 2023; https://arxiv.org/abs/2310.16028)
>
> [9] Feng et al, Towards demystifying the mystery behind chain of thought: A theoretical perspective (arXiv, 2023)
>
> [10] Li et al. Dissecting Chain-of-Thought: Compositionality through In-Context Filtering and Learning (NeurIPS 2023)

---

> ### Author Response · Authors · 2023-11-20
> **Author response part 3**
>
> > **Comment**:  The definition of hierarchical graph and random graph is confusing to me. It seems possible to convert the random graph in fig. 10 into a hierarchical graph by simply re-grouping the nodes.
>
> **Response**:
> In a hierarchical graph, nodes are pre-partitioned into layers. The only connections allowed are between successive layers and these are chosen randomly. Whereas in a random graph, all connections are chosen at random. It is not possible to simply regroup nodes in a random graph and convert it to a hierarchical graph. To provide evidence for this, we now refer the reviewer to Appendix Figure 9 from our paper, where we highlight and discuss the differences between a hierarchical and random graph. In that figure, the *bottom panels* show the distribution of path lengths in the two families of graphs: hierarchical graphs have an exponential distribution owing to their structure and random graphs have a gaussian distribution. For example, given any pair of nodes, in a hierarchical graph every path between them is of the same length (determined by the difference of their layer numbers), whereas in a random graph there is a diversity of path lengths (for instance, there might be an edge between the nodes as well as a path of length more than 10, as is the case with our synthetic random graphs). We wish to emphasize that our specific approach to using a hierarchical graph is designed to enable more precise control over the data distribution.
>
> To further ensure this point is clear, we have added pseudocode for the algorithms for generating both the random and hierarchical graphs—**Please refer to Algorithms 1 and 2 in the Appendix A.3**. We hope that this,along  with the path characterization provided in Figure 9, addresses concerns about possible similarities between the two setups.
>
> > **Comment:**
> The highlighted finding, i.e., the step-wise inference gap is influenced by (i) the underlying DAG structure, and (ii) the length of the training samples is only studied in a very shallow level. The DAG structure in the context seems to be not well-defined and well-categorized. The discussion about the length of the training samples mainly focuses on how it affects the length of the model output.
>
> **Response:**
> Our highlighted findings in a single graph scenario include: (i) the inference gap consistently persists across diverse data distributions in the training data (Figure 3); (ii) a diversity versus accuracy trade-off exists in finite temperature stepwise inference for transformers (Figure 4); and (iii) demonstrating that there is a bias towards shorter paths and the observation that over the course of training the model first learns to take correct steps and then acquires the capability of global planning (Figure 5). We recognize that our approach serves as a first step towards formulating a set of hypotheses for further validation in large-scale, realistic settings, and there is considerable scope for extending our research to more realistic settings such as different data distributions, noisy or incomplete information, and different graph structures. As part of this endeavor, We have now repeated our analysis on a setup where a ground-truth graph is “noisily observed”. Specifically, every time a path is sampled between two nodes, we intentionally corrupt a random fraction of the tokens on the path by 5-20%. The model is now trained on these corrupted paths—this randomization further ensures that overfitting is infeasible, since a given path is never seen twice (due to corruptions). *In this setup, we again see that our claim on the performance gap induced by stepwise inference continues to persist.* We have also defined our training protocol and description of the graphs used in training in more detail (in Appendix).
>
>
> Summary: We appreciate your constructive feedback, which helped us to clarify the motivation, objectives, and limitations of our synthetic data approach. We hope that the contributions of our work are now clearer and that the reviewer will consider raising their score to support the acceptance of our work.

---

> > ### Comment · Reviewer_3PuG · 2023-11-22
> >
> > Thanks for the authors' point-to-point and detailed responses. I appreciate the efforts they made to address my concerns. However, my major concerns remain unchanged. Solely being able to "reproduce an array of phenomena observed in LLMs" on the **synthetic graph data** does not serve as strong evidence that the current findings are useful for LLMs in real-world tasks. An interesting aspect of the step-wise inference process of LLMs is that it naturally extracts a DAG-like structure from the **raw-data** (featuring noisy, not-that-structured often distributed observations). However, directly feeding the already highly-structured clean graph data into the model completely skip this abstraction step. In addition, the random graph and hierarchical graph definition are still not well-established given the explanations. As long as there is no loop in the graph, it is trivial to convert the "random graph" into the "hierarchical graph" defined in this paper. To be more specific, it is clear that a hierarchical DAG generated using algo 2. in the appendix can also be generated by algo 1, by simply re-ordering the nodes in the `adjMatrix` in algo 1.
> >
> > Therefore, as the authors also acknowledge, the current "proposed hypotheses are not conclusive about the mechanisms underlying LLMs" given the insufficient experiment evaluation and/or modeling of the step-wise inference process. My suggestion is that the authors can consider reproducing their findings using at least synthetic language datasets (as the referred works [1,2,3,4] did) and reformulate the model to make a stronger, clearer and more serious connection with the step-wise inference process of LLMs. Considering the author responses and comments from other reviewers, I will keep my rating as it is and stick to rejecting this work.

---

> > > ### Author Response · Authors · 2023-11-22
> > >
> > > Dear Reviewer,
> > >
> > > We thank you for the engagement with our rebuttal.
> > >
> > > As we have detailed in our responses, in addition to reproducing an array of phenomena to confirm our synthetic setup, our main contributions are *formulating novel mechanistic hypotheses*. To be clear, the reviewer is of the opinion that these mechanistic hypotheses are not enough to justify publication of our work, until they are tested in more realistic LLMs.
> > >
> > > In fact, it is more of a philosophical debate as to whether “formulating mechanistic hypotheses” is itself worthy of publication. Indeed, traditionally, the field of machine learning has required thorough benchmarking in realistic settings (e.g., ImageNet for computer vision) to validate the engineering value of the work. However, we argue that as generative AI is rapidly and widely adopted in society, there is an increasing need to better “understand” these complex systems using a “modeling” approach from the natural sciences. In the natural sciences, such as physics or computational neuroscience, we often have “modeling” work whose only contribution is to formulate a simple model and propose mechanistic hypotheses.
> > >
> > > Here, we show through examples that the deep learning research community is rapidly building consensus on the value of this scientific modeling approach, which is similar in spirit and style to ours. For example, Prystawski et al [1] study very simple linear chain graphs theoretically to explain the ‘reasoning gap’. Their contribution was to provide the following mechanistic hypothesis: “The stepwise reasoning gap arises due to the local nature of the training data”. This was followed by simulations on synthetic bayes networks. This paper is now a NeurIPS 2023 oral. Another example is Li et al [2] who study how transformers learn different MLPs in context. Their main contribution was the following mechanistic hypothesis: “Chain-of-thought improves accuracy and sample efficiency by decomposing problems into 2 subtasks: filtering, followed by in-context learning”. This paper is also set to appear at NeurIPS 2023. We believe these mechanistic hypotheses are invaluable in the understanding of LLMs at scale, despite these studies not performing any large scale experiments.
> > >
> > > We think understanding stepwise inference is a grand challenge for LLM research and it is unreasonable to expect that a single study can both design synthetic tasks and experiments and formulate mechanistic hypotheses and test them in large-scale models.
> > >
> > > [1] Eric Prystawski et al. Why think step by step? Reasoning emerges from the locality of experience (https://nips.cc/virtual/2023/oral/73821)
> > >
> > > [2] Yincong Li et al. Dissecting Chain-of-Thought: Compositionality through In-Context Filtering and Learning (https://openreview.net/forum?id=xEhKwsqxMa)

---

> > > > ### Comment · Reviewer_3PuG · 2023-11-22
> > > >
> > > > Thanks for your follow-up discussion and pointers to these related works, which are very helpful indeed. First of all, let's be very clear that I'm not against papers that focus on scientific modeling. But can the authors be more explicit and specific about what is the theoretical contribution in the current modeling of step-wise inference process of LLMs (other than the direct application of the definition of DAGs)? The referred work [1] and [2] did study the related processes using simplied models and datasets, but also provided enough insights using well-established theoretical frameworks and convincing derivations. This, to my opinion, is a key part for scientific modeling, i.e., not only do we make guesses, but the insights and discoveries derived from the hypotheses are also important. Sometimes, these are more important because we can check whether the modeling is right or wrong from the deduction.
> > > >
> > > > Second, as stated in my response, I'm not against synthetic data. I suggested using at least synthetic **language** datasets as used in the referred works, as directly feeding the already highly-structured clean graph data into the model creates a non-negligible gap between the inputs of the current framework and those actually used for LLMs, which are, as stated, "noisy, not-that-structured with the information distributed among different observations". The emergence of the DAG structure itself in the step-wise inference process is non-trivial and should be more carefully considered.

---

> ### Author Response · Authors · 2023-11-23
>
> We thank the reviewer for continued engagement.
>
> > I suggested using at least synthetic language datasets as used in the referred works,
>
> Firstly, we'd like to point out that [1] and [2] did not use any language datasets in their proofs or experiments. [1] uses a similar setup to ours where tokens correspond to nodes on a graph and [2] uses random 4-64 dimensional euclidean vectors.
>
> Secondly, while [1] and [2] have a fraction of theory results, a majority of synthetic data work does not. As examples from diverse areas, consider: [3] which studies how transformers navigate small tree-structured graphs from CFGs, [4] which studies a synthetic image dataset in diffusion models and empirically shows the delayed emergence of compositionality and [5] which studies RL agents interacting with simple causal DAGs with 5 nodes, with purely empirical results and no theorems with proofs. These are most similar in style to our work, which studies "empirical models" , first validates their approach by reproducing phenomena and then produces hypotheses to test at scale.
>
> Our explicit list of hypotheses and contributions is given at the end of the introduction, just before discussion of our limitations and each hypothesis is stated in the corresponding results section. To summarize everything together:
>
> - We find that the Step-by-Step Inference gap depends on two key factors, firstly the structure of the underlying DAG: the Step-by-Step Inference gap is larger for graphs that are hierarchical as opposed to random and secondly, the length of the training samples: the Step-by-Step Inference gap is larger if the model has been trained on a set of shorter paths that have to be ``stitched" together to build the path during evaluation.
>
> - We find that models have a shorter path bias in the case of high path diversity.
>
> - We systematically and quantitatively characterize the diversity-accuracy tradeoff, showing curves with loss of accuracy and increase of diversity as a function of sampling temperature.
>
> - Over the course of training, we find that global planning capabilities arise after the model has learnt to take correct steps
>
> - We hypothesize that for chain-of-thought and related methods at scale: the model will fail to generalize to reasoning chains longer than those present in its training data.
>
> - When presented with 2 conflicting chains in-context, we find that the model has a strong bias toward choosing the first chain over the second.
>
> [3] Allen-Zhu et al. Physics of Language Models: Part 1, Context-Free Grammar https://arxiv.org/abs/2305.13673
>
> [4] Maya Okawa et al. Compositional Abilities Emerge Multiplicatively: Exploring Diffusion Models on a Synthetic Task. NeurIPS 2023 (https://arxiv.org/pdf/2310.09336.pdf)
>
> [5] Ishita Dasgupta et al. Causal Reasoning from Meta-reinforcement Learning ICLR 2019 (https://arxiv.org/abs/1901.08162)

---

> ### Comment · Reviewer_3PuG · 2023-11-23
>
> Thank the authors for their efforts to continue this discussion.
>
> I would like to again point out that first, my suggestion about using synthetic language datasets originates from the following two concerns: **i)** the lack of significant theoretical contribution other than the direct application of the definition of DAGs (if we were to verify the hypotheses on synthetic graph data as in [1, 2]), and **ii)** in addition to i), the non-negligible gap between the inputs of the current framework and those actually used for LLMs, i.e., insufficient empirical evidence for safely transferring the current findings to real-world tasks. These two concerns together undermine my confidence of the current submission being acceptable to a top conference like ICLR.
>
> I thank the authors for further pointing me to [3], [4] and [5]. While [3] is surely an interesting piece of work using only synthetic data with few or no theoretical deduction, the hypotheses are examined with extensive, comprehensive and well-designed experiments which I highly appreciate. These experiments are scaled up and deliberated designed to closely approximate the richness of real language (e.g., see Sec. 2.2 Why Such CFGs in [3]) unlike those in the current submission. [4] and [5], however, are somehow unrelated to the current topic of our discussion, in my opinion. Although [4] did use synthetic data to probe the compositional generalization ability of conditional diffusion models as the core motivating examples and results, there is a section, i.e., `Sec. 4.2 Additional Experimental Results with Real Data' to showcase that the observations are truly consistent (at least partially) with real data. I am admittedly not very familiar with the area of causal reasoning, which is the case in [5] and the case is seemingly not very related to the current problem.
>
> Again, I appreciate the efforts the authors made to try to address my concerns and think the idea of modeling step-wise inference as graph navigation is interesting. It is not likely, however, that I will change my opinion with more listed related/unrelated works if the main concerns, which I believe are clearly stated, are not addressed. I respect the endeavors and opinions from other reviewers, and honor the final decision made by the committee no matter what that will be.

---

### Official Review · Reviewer_7XTH · 2023-11-05

**Soundness:** 2 fair
**Presentation:** 3 good
**Contribution:** 2 fair
**Rating:** 6
**Confidence:** 2

**Summary:**

This paper introduces and explores a novel paradigm that casts stepwise inference, a crucial element in logical reasoning, as a graph navigation challenge. Using directed acyclic graphs (DAGs) inspired by computational graphs and execution traces, the paper proposes a synthetic autoregressive language model setup for solving navigation tasks. The primary aim is to simplify, control, and interpret the mechanisms behind stepwise inference in Large Language Models (LLMs). This work serves as a foundational step towards creating a controllable and interpretable data generation process, offering insights into the stepwise inference in autoregressive transformers, which can inspire future research in logical reasoning and stepwise inference.

**Strengths:**

1. This paper presents an innovative framework that leverages stepwise inference within transformer models to navigate complex graph structures, showcasing a significant advance in understanding logical reasoning paths. The method enhances the interpretability of models but with a deeper mechanistic insight.
2. The paper integrates theoretical concepts with empirical validation, showcasing a comprehensive study on synthetic graph navigation tasks. Experimental design and results, particularly the diversity vs. accuracy trade-off, provide a compelling case for the model's efficacy and reliability.
3. The paper also introduces a data generating process that augments the richness of training datasets, allowing for more robust model training.

**Weaknesses:**

1. While the paper claims to address stepwise inference in transformers and introduces a graph navigation model, the experiments seem to focus narrowly on synthetic tasks without sufficient evidence of the model's generalizability to a more realistic or applicable datasets.
2. The approach primarily involves modeling the decision-making process using directed acyclic graphs (DAGs). However, there are concerns that the model may overfit these synthetic graph structures. The paper does not adequately address how the model handles noisy real-world graphs, which may exhibit cycles, incomplete information, or random behavior that are common in real-world applications. In this context, a deeper analysis of the robustness of the proposed method is crucial.

**Questions:**

Illustrated in the weaknesses.

---

> ### Author Response · Authors · 2023-11-20
> **Author response Part 1**
>
> Thank you for your detailed review and positive feedback! We are glad that you think our paper presents “an innovative framework”, showcases a “significant advance in understanding logical reasoning paths” and that our study was “comprehensive”. We were also excited to study the diversity-accuracy tradeoff, and believe ours is the first work to systematically and quantitatively study this phenomena. Our goal was exactly this: to devise and explore an appropriate systematic synthetic framework to study the phenomenology of stepwise inference. We address your specific questions below.
>
>
> > **Comment:** While the paper claims to address stepwise inference …, the experiments seem to focus narrowly on synthetic tasks without sufficient evidence of the model's generalizability to a more realistic or applicable datasets.
>
> **Response:** Thank you for raising this comment. We have substantially revised our manuscript to address your concern by further motivating the explicit connection between graph navigation and stepwise inference in more realistic settings. Specifically, we have rewritten parts of the introduction (we have elaborated our exposition on the relation between stepwise inference and graph navigation, added a discussion of the limitations of a synthetic data approach), contextualized our results more thoroughly with existing literature, and also edited the formatting and arrangement of figures and results in paper (including promoting the training dynamics figure to the main text since it shows that global planning capability arises *after* models learn to take correct steps). As an example, our edits now highlight several recent studies that either use or model stepwise inference with LLMs to study tasks that are specific instantiations of our proposed graph navigation problem (e.g., see [1, 2, 3, 4]). We have also included a figure in Appendix (Fig. 9) that describes example prompts and model outputs from each of these papers to exemplify the relation of our proposed synthetic benchmark to realistic setups and tasks considered by prior work in showing the value of stepwise inference. Given these (very recent) studies, we argue the connection between stepwise inference, planning, and graph navigation is explicit and our updated draft ensures this connection is emphasized upon.
>
> We’d also like to contextualize how our methodology of proposing and studying a synthetic problem fits into the literature. Specifically, the goal of our work was to analyze an interesting phenomenon witnessed in LLMs, i.e., stepwise inference, by designing an abstraction of the problem. This enables development of precise mechanistic hypotheses. In LLM and generative model literature, this is a fairly common approach. For example, [5,6] use a synthetically designed setup to develop hypotheses for how ``knowledge” about an entity is stored in pretrained model, show that such knowledge can be manipulated via relatively simple linear transformations. [7] used a procedurally defined synthetic dataset with a relatively small model to demonstrate that emergent abilities seen in neural networks are partially driven by the compositional nature of real world data. In other work [8], use Tracr-compiled transformers to show that if primitive operations involved in a formal algorithm can be implemented in a model, stepwise inference is sufficient for length generalization. Similarly, [9] used context-free grammars to demonstrate that stepwise inference is sufficient to solve problems that require dynamic programming. [10] use CoT to show transformers can learn arbitrary compositions of MLPs in-context. Our work lies in the same regime as these studies and has the goal of defining a well-designed task for studying stepwise inference protocols.
>
>
> [1] Mommenejad et al. Evaluating Cognitive Maps in Large Language Models with CogEval: No Emergent Planning (NeurIPS 2023)
>
> [2] Saparov et al. Language Models Are Greedy Reasoners: A Systematic Formal Analysis of Chain-of-Thought (ICLR 2023)
>
> [3] Dziri et al. Faith and Fate: Limits of Transformers on Compositionality (NeurIPS 2023)
>
> [4] Hou et al. Towards a Mechanistic Interpretation of Multi-Step Reasoning Capabilities of Language Models. (arXiv, 2023)
>
> [5] Zeyuan Allen-Zhu and Yuanzhi Li, (2023) Physics of Language Models: Part 3.1, Knowledge Storage (2023)
>
> [6] Zeyuan Allen-Zhu and Yuanzhi Li, Physics of Language Models: Part 3.2, Knowledge Manipulation (2023)
>
> [7] Maya Okawa et al. Compositional Abilities Emerge Multiplicatively: Exploring Diffusion Models on a Synthetic Task (NeurIPS 2023)
>
> [8] Zhou et al. What Algorithms can Transformers Learn? A Study in Length Generalization. (arXiv, 2023; https://arxiv.org/abs/2310.16028)
>
> [9] Feng et al, Towards demystifying the mystery behind chain of thought: A theoretical perspective (arXiv, 2023)
>
> [10] Li et al. Dissecting Chain-of-Thought: Compositionality through In-Context Filtering and Learning (NeurIPS 2023)

---

> > ### Author Response · Authors · 2023-11-20
> > **Author response Part 2**
> >
> > > **Comment:** The approach primarily involves … there are concerns that the model may overfit these synthetic graph structures. The paper does not adequately address how the model handles noisy real-world graphs, which may exhibit cycles, incomplete information, or random behavior that are common in real-world applications.
> > **Response:** These are valid concerns and we have both highlighted our existing results and added a new experiment to address them, as discussed below.
> >
> > **We have added new experiments with corrupted graphs.** Thanks to your feedback, we have added a new experiment to better model noisy graphs. In the experiment, we have now repeated our analysis on a setup where a ground-truth graph is "noisily observed". Specifically, every time a path is sampled between two nodes, we intentionally corrupt a random fraction of the tokens on the path by 5-20%. The model is now trained on these corrupted paths—this randomization further ensures that overfitting is infeasible, since a given path is never seen twice (due to corruptions). *In this setup, we again see that our claim on the performance gap induced by stepwise inference continues to persist.*
> >
> > **Test-train split and randomness by sampling from diverse paths:** We’d also like to clarify and emphasize that **overfitting cannot occur in our setup**. In our current setup, please note that given a start and goal node pair, there is a huge diversity of paths between them: for example, please see Appendix Figure 9, which shows that there are up to $10^3$ valid paths between a pair of nodes. We divide all node pairs into test and train.  For the train node pairs, in each context window receives 1 random path:  this itself is a source of randomness that we now highlight better in the revised manuscript. This randomness corresponds to how a particular stepwise path is sampled in “noisy” real-world settings. (e.g., a particular sequence of logics out of several possible solutions is sampled in a solution set of math word problems.) Importantly, this randomness also acts as a strong regularizer against overfitting on the training set. Further, all our results are on the test set node pairs that have never been seen in a train set, i.e., overfitting to this data is not possible. We note we have updated discussion of our training protocol in Appendix A.3 to summarize these details better.
> >
> > Summary: Thank you very much for your constructive feedback! We hope that we have sufficiently answered your questions with the above new experimental results, arguments and justifications, and hope that you will consider increasing your score to support the acceptance of our work.

---

### Author Response · Authors · 2023-11-20
**Global Response: Motivation and limitations of our synthetic task approach**

We thank the reviewers for their thoughtful feedback and for providing us the opportunity to clarify the contributions of our approach. Below, we further discuss the motivation and limitations of our work.

In our study, we aimed to explore the mechanisms that improve the performance of auto-regressively trained transformers through stepwise inference. To this end, we adopted a methodology analogous to the **model-experimental-systems approach** prevalent in the natural sciences, in which simplified, controllable, and steerable experimental systems are studied to develop mechanistic hypotheses. We can answer questions like: what are the right intuitions and conceptual frameworks and what tooling and techniques do and do not work when one wants to analyze models and model behavior. Can we predict, a priori, what tools and techniques are useful and what “knobs” in the data can we turn to control a phenomena of interest? Our approach began by establishing an empirical foundation by replicating phenomena observed in LLMs, such as the stepwise inference gap, in a controlled graph navigation task. Leveraging the tractability of our model, in the single graph scenario, we were able to uncover a **path stitching** mechanism that contributes to generalization, and we quantified a bias within transformers that favors shorter path solutions. In addition, our model allowed us to precisely characterize the **diversity versus accuracy trade-off** observed in finite temperature stepwise inference. Further, we also discovered a **short path bias** in stepwise inference protocols. We believe that these novel mechanistic hypotheses formulated within our synthetic experimental framework are valuable contributions to the field. *They provide a fundamental understanding that can guide further research into the intricate behavior of auto-regressively trained Transformers.* However, we fully recognize and acknowledge in our manuscript that additional studies are needed to validate these hypotheses robustly and at a larger scale typical of real-world LLM scenarios. To address this, we have added a new paragraph to the paper (see end of Sec. 1, Introduction) that carefully outlines the limitations of our approach. This includes an acknowledgement of the simplified nature of our setup and a discussion of the differences in scale and data diversity between our model and real-world LLMs.

---

### Meta-Review · Area_Chair_W36e · 2023-12-12

**Metareview:**

This paper studies the stepwise inference problem in LLMs by formulating the problem as an abstract graph navigation problem. By this, the paper seeks to interpret the mechanisms behind stepwise inference in LLMs. All the reviewers agree that the problem studied is significant and is foundational to understand transformer-based LLMs. However, the main concern raised by the reviewers is that the work is restricted to study of synthetic tasks, which makes it hard to judge whether the results and conclusion can be generalized to more realistic problems. In practice, LLMs are generally pretrained on massive amount of text data, and then finetuned to demonstrate strong out-of-distribution generalization capabilities. Besides one concern mentioned by Reviewer 3PuG about the natural language ambiguity and variance, one other concern is that not all of the training data are in the form of graph navigation style, and the situation is more complicated than the simplified setting here. Therefore, making strong conclusions about LLM reasoning capability based on the simplified setting may be a bit risky. However, the paper does perform some interesting studies with potentially useful takeaways. The authors are encouraged to submit to future venue by continuing the investigation and possibly extending the experimental study further to additional more realistic formulations.

**Justification For Why Not Higher Score:**

The results and conclusions of the paper are derived based on synthetic tasks, which makes it harder to generalize to the situation in real LLM problems.

**Justification For Why Not Lower Score:**

N/A

---

### Decision · Program_Chairs · 2024-01-16

Reject